# Microstructure and Mechanical Properties of Refill Friction Stir Spot-Welded Joints of 2A12Al and 7B04Al: Effects of Tool Size and Welding Parameters

**DOI:** 10.3390/ma17030716

**Published:** 2024-02-02

**Authors:** Yisong Wang, Pengyang Li, Haitao Jiang, Kang Yang, Zhenhao Chen, Haijiao Chuai, Xiaoyan Wu, Qiang Meng, Lin Ma

**Affiliations:** 1National Engineering Research Center for Advanced Rolling and Intelligent Manufacturing, University of Science and Technology Beijing, Beijing 100083, China; wangys080@avic.com (Y.W.); wuxiaoyan@ustb.edu.cn (X.W.); 2AVIC Manufacturing Technology Institute, Beijing 100024, China; mengq009@avic.com; 3College of Aerospace Engineering, Shenyang Aerospace University, Shenyang 110136, China; pyli@email.sau.edu.cn (P.L.); ykangok@sau.edu.cn (K.Y.); 13243907917@sau.edu.cn (Z.C.); 18840617461@sau.edu.cn (H.C.); 4College of Material Science and Engineering, Shenyang Aerospace University, Shenyang 110136, China

**Keywords:** refill friction stir spot welding, welding of dissimilar aluminum alloys, effective welding depth, fracture behavior

## Abstract

To solve problems in dissimilarly light metal joints, refilled friction stir spot welding (RFSSW) is proposed instead of resistance spot welding. However, rotation speed, dwell time, plunge depth, and the diameter of welding tools all have a great influence on joints, which brings great challenges in optimizing welding parameters to ensure their mechanical properties. In this study, the 1.5 mm thick 2A12Al and 2 mm thick 7B04Al lap joints were prepared by Taguchi orthogonal experiment design and RFSSW. The welding tool (shoulder) diameters were 5 mm and 7 mm, respectively. The macro/microstructures of the cross-section, the geometrical characteristics of the effective welding depth (EWD), the stir zone area (SZA), and the stir zone volume (SZV) were characterized. The shear strength and failure mode of the lap joint were analyzed using an optical microscope. It was found that EWD, SZA, and SZV had a good correlation with tensile–shear force. The optimal welding parameters of 5 mm diameter joints are 1500 rpm of rotation speed, 2.5 mm of plunge depth, and 0 s of dwell time, which for 7 mm joints are 1200 rpm, 1.5 mm, and 2 s. The tensile–shear force of 5 mm and 7 mm joints welded with these optical parameters was 4965 N and 5920 N, respectively. At the same time, the 5 mm diameter joints had better strength and strength stability.

## 1. Introduction

In aviation, aerospace, and other manufacturing fields, developing lightweight products is a growing trend. The use of 2xxx and 7xxx aluminum alloys with light weight and high strength is one of the most effective measures to meet this requirement [1]. However, joining aluminum alloy parts is a challenge for manufacturing operations. Resistance spot welding is widely used to join aluminum alloys. However, there are disadvantages such as large deformation, high energy consumption, inconsistent welding quality, and frequent electrode loss in resistance spot welding [2]. Therefore, it is of great significance to promote the use of lightweight component manufacturing using new technologies for joining aluminum alloys.

Refill friction stir spot welding (RFSSW) is a green, energy-saving, environmentally friendly, and efficient spot-welding technology developed based on friction stir welding [3,4,5]. The technology is similar to friction stir welding, in which a solid joint is formed while the workpiece is kept below the melting temperature under the action of frictional heat and mechanical forces [6,7]. RFSSW has been identified as having better application potential for aluminum alloy joining with no exit hole after welding (and smooth surfaces), which overcomes the disadvantages encountered in resistance spot welding of aluminum alloy [8,9,10]. The friction heat generated by the rotating tool in area of contact with the specimen surface causes the material to flow in the stir zone (SZ) [11]. Welding parameters, such as tool rotation speed (RS), dwell time (DT), axial force, and plunge depth (PD), usually affect the flow and distribution of material grains [12,13,14]. The flow of the material and the resulting microstructure greatly influence the defects and pores, which often compromise the physical properties and performance characteristics of welded joints [6,15,16,17].

Among so many welding parameters that affect the microstructure and mechanical properties of RFSSW joints, it is still unclear which are the most important. For example, Li et al. reported that the plunge speed of the RFSSW process was affecting the mechanical properties of the 2A12Al-T42 joints [18]. However, Yu et al. found that the PD directly affected the microstructure and mechanical properties of Al/steel joints [19]. In addition, Fritsche et al. found that the mechanical properties and morphology of AlSi10Mg and 7075Al-T6 joints were optimal at medium RS (1500 rpm) [20].

For RFSSW, the size of the welding tool also affects the joint size as well as the microstructure and mechanical properties of the joints. Among them, differently sized (5–9 mm) diameter welding tools, especially 9 mm tools, are commonly used [21]. According to relevant research results, small RFSSW joints have better mechanical properties [12,22,23]. However, it is not known whether a smaller weld diameter can improve the mechanical properties of the joint.

No matter whether using the same or different aluminum alloy, the relationship between defects and the mechanical properties of joints can be determined by optimized welding parameters and the selection of welding tool size [24]. However, for the RFSSW of dissimilar aluminum alloys, the mechanical properties are different due to the different initial compositions [25]. Therefore, appropriate welding and tool parameters are very important for obtaining excellent structure and performance of dissimilar aluminum alloy joints. At present, there is scarce research on the parameter optimization of dissimilar aluminum alloy joints and the study of microstructure-related mechanical properties of different types of welded joints.

A welding tool with optimized diameters can produce excellent welding blocks, which increase the bearing capacity of a joint. Therefore, in this study, welding tools with different diameters were used to join 2A12Al and 7B04Al, and the microstructures and mechanical properties of the joints were compared under the conditions of different RSs, PDs, and DTs. The microstructure-related mechanical properties were studied and the fracture behavior was described.

## 2. Materials and Methods

### 2.1. Refill Friction Stir Spot Welding

Welding materials 2A12Al-T42 and 7B04Al-T74 were selected with thicknesses of 1.5 mm and 2.0 mm, respectively. The compositions of aluminum alloys are shown in Table 1.

Movable gantry refill friction stir spot welding equipment (FSSW-DLM-204, CFSW Co., Ltd., Beijing, China) was used to weld alloys. For conventional welding tools, as shown in Figure 1a, the diameters of the SZ were 5 and 7 mm. For the small-sized welding tool, the diameters of the clamping ring and shoulder were 15 mm and 5 mm. The diameter of the pin was 2.5 mm. For the large-sized welding tool, the diameter of the clamping ring was 18 mm. The diameters of the shoulder and pin were 7 mm and 4 mm. Both the pin and shoulder were threaded on the outside. The PDs were set at 1.5, 2, and, 2.5 mm. The RSs were 900, 1200, and 1500 rpm, with DTs of 0, 1, and 2 s, respectively. There were nine experimental settings of welding parameters based on the Taguchi method using Minitab18 software (https://www.minitab.com/), as shown in Table 2 [26,27,28].

The welding process was divided into four stages, as shown in Figure 1b. The first stage started from the contact between the rotating tool and the top surface of the 2A12Al-T42 upper plate to preheat the alloys. After preheating, the shoulder was plunged at a rate of 50 mm/min until contact with the top surface of the 7B04Al-T74 bottom plate. This stage led to material agitation and plastic flow, which was prone to generate friction heat in the stirring pin. The stirring pin retracted to create a chamber for the displaced material to flow into. The next stage was to plunge the stirring pin into the upper plate at a rate of 240 mm/min, leading to a positive mixing of the material in the upper and bottom plates. The tool’s plunge helped increase the volume of the stirred material. In the whole process of welding, the moving speed ratio of the shoulder to the pin was 1:3. In the next stage, the cross-sections of the stirring pin and shoulder were at the same level, thus greatly increasing the stirring area and the generation of friction heat; then, the welding was complete.

### 2.2. Microstructure and Mechanical Characterization

The microstructure of welded specimens of dissimilar aluminum alloys was analyzed using an optical microscope (OM, OLYMPUS-GX71, Olympus IMS, Houston, TX, USA) and scanning electron microscope (SEM, CIQTEK-SEM3100, 20 kV, CIQTEK, Hefei, China). The material flow and fracture evolution in the mixed zone of the welded samples were evaluated. The length of the welded sample was 175 mm and the thickness was 3.5 mm. The initial surface treatment was carried out with sandpaper of different grit sizes, and then the surface was polished with an alumina solution. The specimen surface was etched using the Keller solution for 5 s before observation.

The tensile–shear properties of welded specimens were tested with an Instron 8801 (ASTM E8 standard [29]) at a rate of 2 mm/min, as shown in Figure 1c. Three samples were selected for each parameter of the joints for mechanical property testing.

## 3. Results and Discussions

### 3.1. Mechanical Property Evolutions

The three factors (RS, PD, and DT) with three levels were selected for the experiment (Table 2), which was conducted using an orthogonal table L9(34) to arrange joint diameters of 5 mm and 7 mm. Figure 2 shows the tensile–shear force (F) of RFSSW joints with different joint diameters (JD) with different welding parameters. All the results of the mechanical properties are listed in Table 3.

When the joint diameter was 5 mm, the tensile–shear force of RFSSW joints increased with the increase in the PD (Figure 2a). From Table 3, it can be observed that the sample of 1500-2.5-0 exhibited the highest tensile–shear force at 4965 N.

Just as shown in Figure 2b, when the joint diameter was 7 mm, the tensile–shear force was more complicated and affected by all three factors. Among these, the sample of 1200-1.5-2 demonstrated the highest tensile–shear force at 5920 N. The corresponding parameters were an RS of 1200 rpm, PD of 1.5 mm, and DT of 2 s. Additionally, it can be observed from Table 3 that the tensile–shear force of the joint with 7 mm diameter was higher than that of the joint with 5 mm diameter.

To remove influences caused by the diameter, the ratio of tensile–shear force to the joint diameter (F/JD) was used as a reference for mechanical properties. The mechanical properties of the joints with a 5 mm diameter were better than those with a 7 mm diameter. The maximum value for a joint with a 5 mm diameter was 993 N/mm, and that for a joint with a 7 mm diameter was 845 N/mm.

The orthogonal experiment results are summarized in Table 4. For joints with a diameter of 5 mm, the range value showed that R_PD_ > R_DT_ > R_RS_, indicating that the rank of influencing factors was PD, DT, and RS, as shown in Figure 3a. While the rank of influencing factors was PD, RS, and DT (Figure 3b) for 7 mm joints.

RS refers to rotation speed, PD refers to plunge depth, DT refers to dwell time, JD refers to joint diameter, and F refers to tensile–shear force.

The heat input of the welded joint is determined by both the RS and DT in the welding process, which also impacts the refill effect of the keyhole. The PD influences the volume of extruded base material during shoulder insertion, as well as the depth of the keyhole and its impact on the lower 7B04Al plate. When pulling back the shoulder, material from the mixing area is refilled into the keyhole. In conclusion, PD has a significant influence on mechanical properties. Thus, for 5 mm of JD, the best welding parameter was 2.5 mm of PD, 0 s of DT, and 1500 rpm of RS, while that was 1.5 mm of PD, 1200 rpm of RS, and 2 s of DT for the 7 mm joints. Further, the tensile–shear force of 5 mm joints was stable.

### 3.2. Effect of Welding Parameters on Microstructure

The microstructures of RFSSW joints can be divided into the stir zone (SZ), thermomechanically affected zone (TMAZ), heat-affected zone (HAZ), and the base metals (BM), as shown in Figure 4 and Figure 5. Besides, there were certain defects such as hooks, holes, and adhesion ligaments, as shown. The formation of hook defects was attributed to external loading.

Figure 4a–c show the cross-sectional morphologies at the RS of 900 rpm. At different PDs (1.5, 2.0, and 2.5 mm), distinct material defects could be observed. As the PD increased, there was an obvious increase in the volume of plastic deformation material along the depth, while the width of the SZ slightly expanded.

When the PD was set at 1.5 mm, hook defects appeared at the interface with a small angle inclination which was formed by external loading. The material flow near the above interface was random, and numerous aluminum-clad layers were distributed specifically within the middle region as well as the interface of SZ and TMAZ.

For the PD of 2.0 mm, the aluminum-clad layer took on a semi-circular shape at the bonding interface. Compared to the joint with the PD of 1.5 mm, the two longitudinal aluminum-clad layers along with the interface of SZ and TMAZ exhibited greater continuity.

In cases wherein the PD reached up to 2.5 mm, holes emerged at the root part near the shoulder’s lowest depth. Although there was some reduction in aluminum-clad layer thickness occurring across the bond interface and the interface of SZ and TMAZ for such conditions, the presence of the interfaces remained highly noticeable.

As shown in Figure 4d, the RS was increased to 1200 rpm. With the increase in RS, the intensity of material flow in the SZ area increased, and the width and depth of the SZ increased as a whole. In the meantime, there was an obvious downward bending trend in hook defects at the interface. In contrast, there was a wide aluminum-clad layer near the lower surface of the interface.

Figure 4e,f show the cross-section of joints with 2 mm and 2.5 mm PD produced by 1500 rpm, respectively. For the 2 mm PD joint, the hook defects at the interface disappeared, and the aluminum-clad layer was mainly distributed in the shape of “X” at the interface. Further, the refill line could still be observed, but there was no aluminum-clad layer. For the 2.5 mm PD joint, the hook defects at the interface tended to bend downward. In addition, the aluminum-clad area at the interface had two layers, one wide and another narrow, and the length of the SZ and TMAZ near the interface increased.

Figure 5a–c show the representative cross-sectional morphologies of 7 mm diameter RFSSW joints at an RS of 900–1500 rpm with a 2 mm PD. When the RS was 900 rpm, the existence of hook defects could be observed, and the aluminum-clad layer at the interface was still mainly distributed in the middle of the lap interface. When the RS was 1200 rpm, the material flow was intense, and the aluminum-clad layer at the interface appeared to turn upward, while the hook-like defects extended to near the upper surface. When the RS was 1500 rpm, the hook defects at the interface were obvious. At this RS, the material flow was more adequate, the distribution area of the aluminum-clad layer increased, and the material flow of SZ and TMAZ near the interface was significantly enhanced.

With the increase in RS, the material flow in the welding core area was intensified, which was more adequate at the lap interface and the interface of SZ and TMAZ. However, when the RS reached 1500 rpm, an upward flow trend appeared in local areas of the interface, and hook defects extended to the surface. As a result, the joint strength was reduced and unstable.

Figure 6a,b show the holes and end point of effective binding in the 900-2.5-2 joint with a 5 mm diameter. There were also obvious defects in the joint. Figure 6c,d show the SZ/TMAZ interface, TMAZ/HAZ interface, and end point of effective binding in the 5 mm joint of 1500-2-2. The above results showed that joints with a better microstructure could be obtained under high RS. Figure 6e,f show the adhesion ligaments and interfaces formed in the joints without obvious defects produced under the optimal welding parameters (a PD of 2 mm, an RS of 1200 rpm, and a DT of 0 s) with a 7 mm diameter tool. The dark area is the intersection zone of the materials.

### 3.3. Effect of EWD and SZ Area on Tensile–Shear Force

Among the three welding parameters, the RS and DT affected the heat input and material flow in the SZ. The higher the RS and the longer the TD was, the higher the heat input and the more intense the material flow was. These resulted in a deeper effective welding depth (EWD), larger SZ area (SZA), and larger SZ volume (SZV), just as listed in Table 5. The effective welded diameter was the distance between the two end points of the effective binding (Figure 6b,d) of the joints. The EWD refers to the distance from the joint surface to the bottom SZ/TMAZ interface. The values of the representative joints are marked in Figure 4 and Figure 5.

For 5 mm diameter joints with a PD of 2 mm, the EWD was 27%, 24%, and 33% deeper than the PD, while these values for 7 mm diameter joints were 21%, 18%, and 26%, respectively. It can also be seen from the increase in EWD that the longer the DT was, the higher the proportion of depth increase was. Therefore, when the same PD was set, the DT had a greater impact on the EWD. Further, this effect decreased as the JD increased. However, under the same RS, the effect of DT on the EWD was less than that of the PD. As shown in Figure 4a–c, even with the increase in DT, the deeper the PD was, the smaller the increment in EWD (46%, 27%, and 15%) was.

In addition, the increments in the SZA (64%, 32%, and 25%) and SZV (78%, 38%, and 36%) of the 5 mm diameter joints decreased with the increase in the PD. When the PD was 2 mm, the increments in the SZA (32%, 30%, and 40%) and SZV (38%, 36%, and 48%) were both closed. For 7 mm diameter joints, the increments in the SZA were 23%, 23%, and 34%, while the increments in the SZV were 26%, 29%, and 42%, respectively. Under the conditions of high RS and long DT, the affected volume of the material in the joint increased.

To sum up, the most important factor influencing the shape of SZ was the PD. For 5 mm joints, the influence of DT on morphology was more prominent than for 7 mm joints, while the effect of RS was less obvious. Further, the welding parameters and tool diameter affected the flow of materials in a certain volume of SZ.

Through the orthogonal experiment, it was found that the most important factor affecting the mechanical properties of the joint was the PD, which determined the EWD, but the other two factors also affected the SZA and SZV.

Table 5 lists the results for the representative samples, giving the EWD, SZA, SZV, and average tensile–shear force for each combination of RS, DT, and PD. Based on the values in Table 5, Figure 7 was drawn, which shows the trends in EWD, SZA, and SZV versus the trend in tensile–shear force. A similar behavior could be observed between them. For the 900-1.5-0 joint of 5 mm diameter, the specimen had the smallest EWD, SZA, and SZV (2.28 mm, 12.33 mm^2^, and 52.38 mm^3^). In contrast, the largest EWD, SZA, and SZV (2.9 mm 15.63 mm^2^, and 66.14 mm^3^) values were found on the 1500-2.5-0 joint. In both cases, the EWD and SZA values conform to the minimum and maximum tensile shear force values for the same combination of parameters (4033 N and 4965 N). The energy input to the welding parameters could be improved in RFSSW, as the aluminum alloy of the upper plate was extruded to the outside of the tools, and the plastic flow area was expanded. Therefore, a large welding diameter and effective welding depth were obtained [8]. In this case, RFSSW joints with different parameters are contrasted in terms of the welding area. Castro also confirmed the same trend on the RFSSW joints of 3.2 mm thick 2198Al sheets [26,30].

### 3.4. Effect of Tool Size on Fracture Behavior

Several researchers reported that the failure of RFSSW joints of aluminum sheets under tensile shear loads resulted in two typical fracture types: through-the-weld and pull-out [31,32,33] fractures. The first one was a shear fracture, in which the nugget remained partially on the bottom sheet [34]. The second was a plug fracture, where the entire weld nugget was left on the bottom sheet [35]. The fracture morphologies of representative joints with 5 mm and 7 mm diameters selected according to their higher tensile–shear force are shown in Figure 8 and Figure 9, respectively. Due to stress concentrated locally after tensile initiation, cracks started at the end point of the effective binding (Figure 6). Then, they propagated along the path of least resistance through weak bonding at the interface.

The fracture cross-sections of all 5 mm joints showed a shear fracture, usually accompanied by nucleation of the crack from the hook region and propagation along the interface between the upper and lower sheets. The joints showed different crack propagation characteristics, which suggested the welding parameters led to different crack propagation mechanisms [36]. In Figure 8, the red circle zones were the crack initiation points, and the red arrows were the crack propagation path. The crack propagation of the 900-2.5-2 joint received little resistance, and the crack propagated along the adhesion ligament until fracture, as shown in Figure 8a. For the 1200-2-0 joint shown in Figure 8b, the crack still propagated along the adhesion ligament, but tended to propagate along the interface of the SZ/TMAZ. When the welding parameters were an RS of 1500 rpm, a PD of 2 mm, and a DT of 2 s, the crack was more complicated and tended to form plug cracks (Figure 8c). The above results show that the weak parts of the 5 mm joints were the bonding interfaces of the two aluminum alloy sheets, and the interfaces were mainly made up of adhesion ligaments. Even if there were hole defects in the SZ/TMAZ interface in the joint, the crack propagation was not affected.

The fracture mode of the 7 mm joints was a plug fracture, and the nuggets were left on the bottom sheet. As shown in Figure 9a, 2A12Al in the 900-2.5-2 joint was distributed on the upper part of the blue dotted line, and 7B04 was located at the SZ/TMAZ interface. After initiation from the effective binding end points on both sides, the crack propagated along the outer side of the SZ/TMAZ interface to form a plug crack up to the upper surface (Figure 9a). Figure 9b shows the fracture morphology of the 1200-2-0 joint with the nugget remaining on the bottom sheet. The crack propagated upward along the SZ/TMAZ interface, reaching the upper surface of the joint and causing failure. In the tensile shear test, the crack originated from the hook tip and usually extended along the cladding or shoulder exit path, and the crack propagated along the cladding with poor strength [37,38]. However, with these welding parameters, there was a significant path change in crack propagation (shown by the arrows on the left in Figure 9b). It can be seen that the cracks first propagated along the shoulder retraction path and finally only propagated along the relatively weak cladding. Complex crack propagation can increase the tensile–shear force of the joints [39,40].

The fracture morphology of the 1500-1.5-1 joint shown in Figure 9c is similar to that of the 900-2.5-2 joint, showing left crack propagation along the SZ/TMAZ interface. The above fracture morphologies and paths indicate that the RS of 1200 rpm was more conducive to the preparation of high-strength joints. The results were also consistent with the experimental results of orthogonal mechanics.

In addition, the welding parameters only affected the crack propagation path, but did not affect the fracture mode. Meanwhile, the diameters of welding tools had a great influence on fracture mode. According to the mechanical properties of the joint, the F/JD values of the 5 mm joints were higher than the values of 7 mm. The above results showed that RFSSW joints with small diameters have better strength and strength stability.

## 4. Conclusions

In summary, the study demonstrated that 2A12Al and 7B04Al can be welded with high quality by adjusting the parameters of refill friction stir spot welding. The optimal welding parameters of joints with different diameters are different. The optimal welding parameters of refill friction stir spot welding joints with a 5 mm diameter are 1500 rpm of rotation speed, 2.5 mm of plunge depth, and 0 s of dwell time; for 7 mm joints, these are 1200 rpm, 1.5 mm, and 2 s. The small diameter (5 mm) joint has better tensile shear force/diameter than the large diameter (7 mm) joint; these values are 993 N/mm and 845 N/mm, respectively. For 5 mm diameter joints, the effective welding depths, SZA, and SZV were positively correlated with the tensile–shear force. In the tensile shear tests and microstructures’ failure tests, two fracture modes were determined: plug fracture mode (of 7 mm joints) and shear fracture mode (of 5 mm joints). Among the joints studied, those with a shear fracture pattern showed higher strength and strength stability.

## Figures and Tables

**Figure 1 materials-17-00716-f001:**
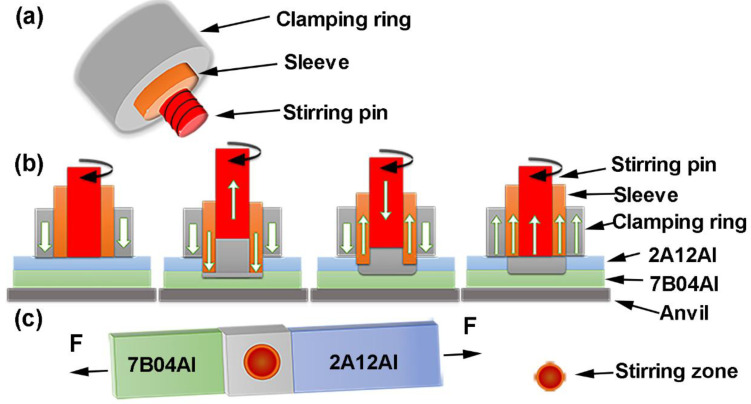
Schematics of the (**a**) welding tools, (**b**) welding process, and (**c**) tensile shear test.

**Figure 2 materials-17-00716-f002:**
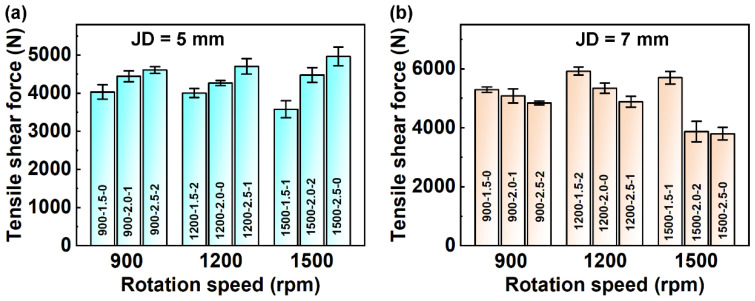
Tensile–shear force of RFSSW joints for various welding parameters with diameters of (**a**) 5 mm and (**b**) 7 mm.

**Figure 3 materials-17-00716-f003:**
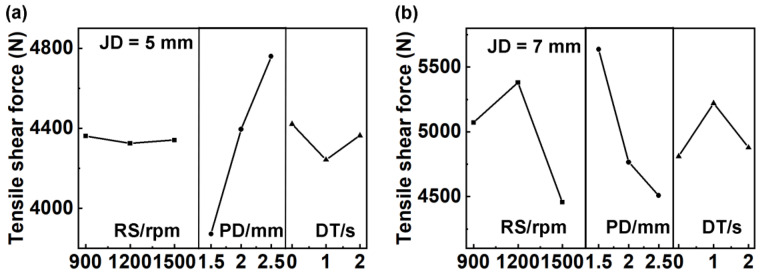
Main benefits of factors affecting mechanical properties of welded joints with diameters of (**a**) 5 mm and (**b**) 7 mm.

**Figure 4 materials-17-00716-f004:**
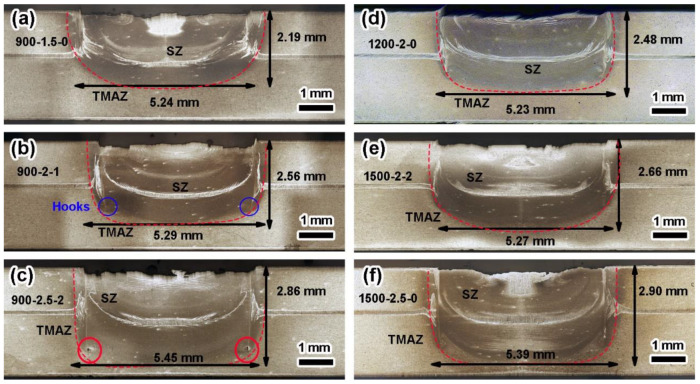
Cross-section morphologies of RFSSW joints under different parameters: (**a**–**c**) at the same RS of 900 rpm with different PDs and DTs; (**d**) at an RS of 1200 rpm, a PD of 2 mm, and a DT of 0 s; (**e**,**f**) at the same RS of 1500 rpm with different PDs and DTs.

**Figure 5 materials-17-00716-f005:**
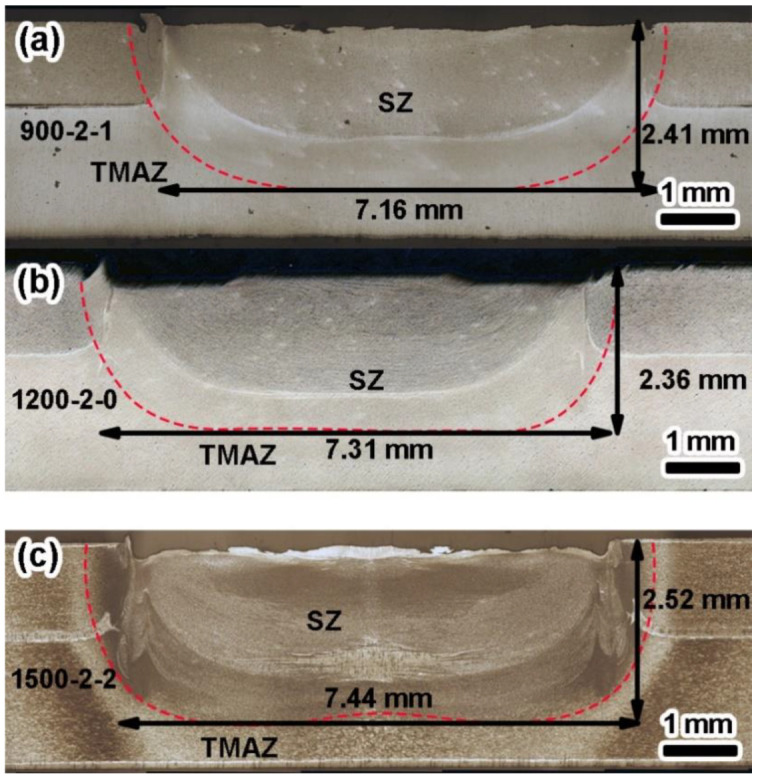
Macroscopic structures of the representative cross-section of joints with 7 mm diameter and 2 mm PD: (**a**) at an RS of 900 rpm and a DT of 1 s; (**b**) at an RS of 1200 rpm and a DT of 0 s; and (**c**) at an RS of 1500 rpm and a DT of 1 s.

**Figure 6 materials-17-00716-f006:**
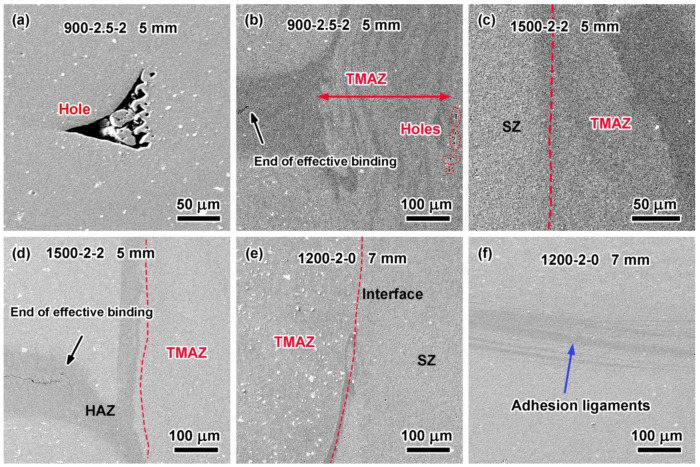
Morphologies of representative RFSSW joints under various welding parameters: (**a**,**b**) hole and end of effective binding in 5 mm 900-2.5-2 joint; (**c**,**d**) interfaces and end of effective binding in 5 mm 1500-2-2 joint; and (**e**,**f**) interface of SZ/TMAZ and adhesion ligaments in 7 mm 1200-2-0 joint.

**Figure 7 materials-17-00716-f007:**
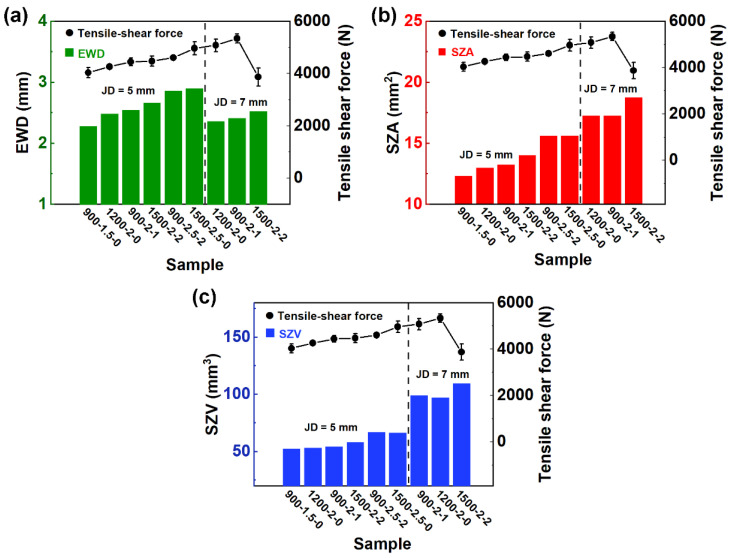
Plots of (**a**) effective welding depth (EWD), (**b**) stir zone area (SZA), and (**c**) stir zone volume (SZV) vs. tensile–shear force mean values for the representative Taguchi array welding condition.

**Figure 8 materials-17-00716-f008:**
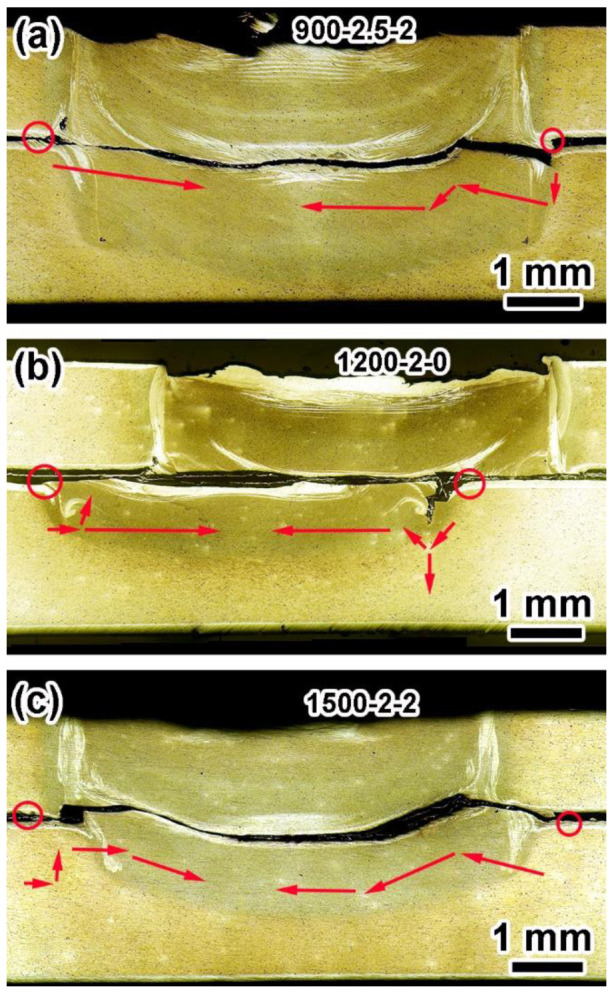
Typical cross-section macrograph of the representative 5 mm joint fractures: (**a**) 900-2.5-2, (**b**) 1200-2-0, and (**c**) 1500-2-2 (red circles and arrows are the points of crack initiation and the directions of crack propagation, respectively.).

**Figure 9 materials-17-00716-f009:**
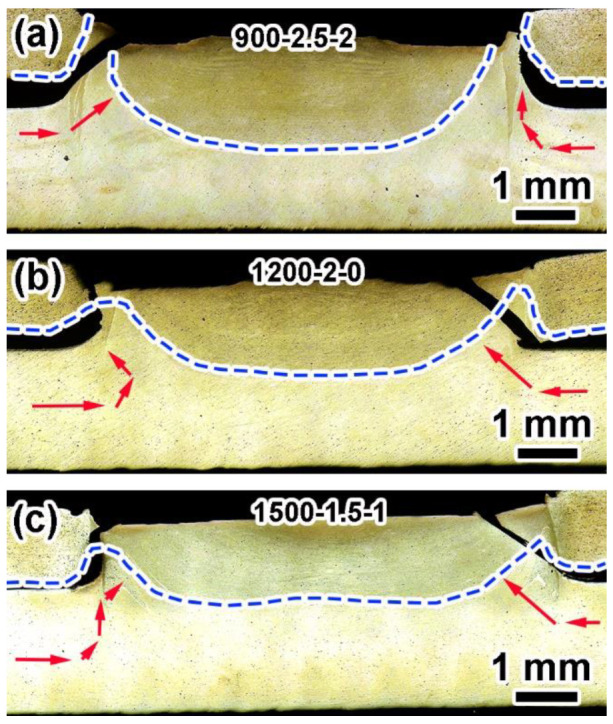
Typical cross-section macrograph of the representative 7 mm joint fractures: (**a**) 900-2.5-2, (**b**) 1200-2-0, and (**c**) 1500-1.5-1 (blue dotted lines and red arrows are the interfaces of alloys and the directions of crack propagation, respectively.).

**Table 1 materials-17-00716-t001:** The compositions of aluminum alloys (wt.%).

Alloy/Element	Al	Cr	Cu	Fe	Mg	Mn	Zn
2A12Al	Balance	--	4.0	--	1.8	0.4	--
7B04Al	Balance	0.1	1.5	0.05	1.8	0.2	5

**Table 2 materials-17-00716-t002:** Optimization of welding parameters based on the Taguchi method.

Sample	Rotation Speed (rpm)	Plunge Depth (mm)	Dwell Time (s)
900-1.5-0	900	1.5	0
900-2.0-1	900	2.0	1
900-2.5-2	900	2.5	2
1200-1.5-2	1200	1.5	2
1200-2.0-0	1200	2.0	0
1200-2.5-1	1200	2.5	1
1500-1.5-1	1500	1.5	1
1500-2.0-2	1500	2.0	2
1500-2.5-0	1500	2.5	0

**Table 3 materials-17-00716-t003:** The values of F and F/JD for the specimens with different JD.

Sample	RS (rpm)	PD (mm)	DT (s)	JD (mm)			
				5		7	
				F (N)	F/JD (N/mm)	F (N)	F/JD (N/mm)
900-1.5-0	900	1.5	0	4033 ± 191	801	5293 ± 94	756
900-2.0-1	900	2.0	1	4444 ± 145	889	5081 ± 244	726
900-2.5-2	900	2.5	2	4608 ± 86	922	4843 ± 64	692
1200-1.5-2	1200	1.5	2	4004 ± 116	801	5920 ± 136	845
1200-2.0-0	1200	2.0	0	4265 ± 70	853	5342 ± 177	763
1200-2.5-1	1200	2.5	1	4705 ± 205	941	4882 ± 187	697
1500-1.5-1	1500	1.5	1	3580 ± 223	716	5699 ± 210	814
1500-2.0-2	1500	2.0	2	4477 ± 196	895	3871 ± 351	553
1500-2.5-0	1500	2.5	0	4965 ± 248	993	3799 ± 210	543

**Table 4 materials-17-00716-t004:** Orthogonal design for process optimization for joints.

JD	5 mm			7 mm		
Factor	RS	PD	DT	RS	PD	DT
K1	4362	3872	4421	5072	5637	4811
K2	4325	4395	4243	5381	4765	5221
K3	4341	4759	4363	4456	4508	4878
Delta	37	834	178	925	1129	409
Rank	3	1	2	2	1	3

**Table 5 materials-17-00716-t005:** EWD, SZA, and SZV measurements for representative specimens.

JD mm	Sample	Diameter of SZ mm	EWD mm	SZA mm^2^	SZV mm^3^	Tensile–Shear Force N
5	900-1.5-0	5.41	2.28	12.33	52.38	4033
	900-2-1	5.21	2.54	13.23	54.12	4444
	900-2.5-2	5.45	2.86	15.59	66.69	4608
	1200-2-0	5.23	2.48	12.97	53.25	4265
	1500-2-2	5.27	2.66	14.02	57.99	4477
	1500-2.5-0	5.39	2.9	15.63	66.14	4965
7	900-2-1	7.16	2.41	17.26	96.99	5081
	1200-2-0	7.31	2.36	17.25	98.99	5342
	1500-2-2	7.44	2.52	18.75	109.50	3871

## Data Availability

Data are contained within the article.

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
