# Peer review of "Microstructure and Mechanical Properties of Refill Friction Stir Spot-Welded Joints of 2A12Al and 7B04Al: Effects of Tool Size and Welding Parameters"

_materials, 2024, doi:10.3390/ma17030716_

Round 1

Reviewer 1 Report

Comments and Suggestions for Authors

Dear authors,

The article entitled "Microstructure and mechanical properties of refill friction stir spot welded joints of 2A12Al and 7B04Al: Effects of tool size and welding parameters" is methodologically well structured, but presents some adjustments that need to be made for publication.

Initially, the abstract does not present the conclusion that the authors reached with the development of the article. Authors must also include the conclusion in the abstract.

The authors carried out a comprehensive and significant literature review, which left the introduction extensive and difficult to understand logically with the objectives of the article. I wish the authors could shorten the introduction, making it more concise and objective.

The methodology is well outlined and the study criteria are well presented. However, there is a lack of details on the equipment used in the research, such as welding equipment. Also, look at the other equipment (including those for Surface Analysis), as they all lack descriptions.

In presenting the results and discussion, the authors did a good job, presenting the data found in the development of the project well. However, I felt the lack of a theoretical framework for discussing the data at some points in this section, such as in the presentation of the results of Effect of welding parameters on microstructure and Effect of EWD and SZ area on tensile shear force.

I strongly suggest that the authors modify the conclusion of the article. The text used as a conclusion to the article is nothing more than results and not a conclusion. What is important in the achievement as a finding for the research carried out? What would be ideal to use as a welding method for 2A12Al and 7B04Al alloys?

Author Response

Dear authors,

The article entitled "Microstructure and mechanical properties of refill friction stir spot welded joints of 2A12Al and 7B04Al: Effects of tool size and welding parameters" is methodologically well structured, but presents some adjustments that need to be made for publication.

Comment 1:

Initially, the abstract does not present the conclusion that the authors reached with the development of the article. Authors must also include the conclusion in the abstract.

Response 1:

Thanks very much for the suggestion. The abstract has been revised. Page 1 lines 16-18 and lines 20-23.

Comment 2:

The authors carried out a comprehensive and significant literature review, which left the introduction extensive and difficult to understand logically with the objectives of the article. I wish the authors could shorten the introduction, making it more concise and objective.

Response 2:

The introduction has been streamlined and revised to make it more logical. Page 1 lines 42-44. Page 2 lines 45-46, lines 48-50, and lines 52-66.

Comment 3:

The methodology is well outlined and the study criteria are well presented. However, there is a lack of details on the equipment used in the research, such as welding equipment. Also, look at the other equipment (including those for Surface Analysis), as they all lack descriptions.

Response 3:

The materials and methods section added subheadings and modified the content in detail. Page 2 lines 84-88 and lines 90-93. Page 3, lines 96-97, lines 99-103, line 107, and lines 115-117. Page 4 line 119.

Comment 4:

In presenting the results and discussion, the authors did a good job, presenting the data found in the development of the project well. However, I felt the lack of a theoretical framework for discussing the data at some points in this section, such as in the presentation of the results of Effect of welding parameters on microstructure and Effect of EWD and SZ area on tensile shear force.

Response 4:

We have modified the results and discussions section. Page 4 lines 122-123, lines 127-128, and lines 133-134. Page 5 lines 146-147 and lines 155-158. Page 6 lines 164-166, lines 169-170, and lines 192-193. Page 7 line 200. Page 8 lines 214-216, lines 219-220, lines 223-224, lines 228-229, line 231, lines 233-234, and lines 236-239. Page 9 lines 241-243, lines 246-250, line 252, and lines 270-274. Page 11 lines 308-309 and lines 311-313.

Comment 5:

I strongly suggest that the authors modify the conclusion of the article. The text used as a conclusion to the article is nothing more than results and not a conclusion. What is important in the achievement as a finding for the research carried out? What would be ideal to use as a welding method for 2A12Al and 7B04Al alloys?

Response 5:

The conclusion has been modified. Page 12 lines 338-349.

Reviewer 2 Report

Comments and Suggestions for Authors

This article reports the evaluation of RFSSW joints of 5 mm and 7 mm diameter of 2A12Al/7B04Al, at different rotation speeds, dwell time and depth of penetration, analyzing the microstructure, mechanical properties, fracture behavior, for joints of 5 and 7 mm diameter; to help determine the optimal welding parameters joints. This submission seems to show interesting results for the purpose of this study, I have minor comments of this document.

Minor coments

-Throughout the writing they use the word sleeve, this word is not appropriate, it is better to change it for shoulder

1. Page 1, 16th line of the abstract: clarify whether the diameters of 5 and 7 mm are from the welding tool or the shoulder or SZ, it is not clear.

2. Page 1, 17th and 18th lines: Statement with incomplete idea “The macro/microstructures of the cross-section, the geometrical characteristics of the effective welding depth (EWD), the stir zone area (SZA), and stir zone volume (SZV)”

3. Page 2, 76th line: Include references in square brackets.

4. Page 2, 85th line and table 1: ¿Is the composition presented in at% or wt%?

5. Page 2, 89th line: clarify whether the diameters of 5 and 7 mm are from the welding tool or the shoulder or SZ, it is not clear. Change the word tools by tool, in singular.

6. Page 2, 91st line: Change the word tools by tool, in singular.

7. Page 2, 90-94th lines: this paragraph should be rewritten because the word “respectively” appears 5 times very close to each other.

8. Page 3, first paragraph: there is no mention that in the first stage occurs initial frictional preheating and for the second stage they don't mention that at the same time that the shoulder penetrates, the stirring pin retracts to create a chamber for the displaced material to flow into.

9. Page 4. In the figure caption the text is repeated, rewrite it.

10. Page 8. Figure 6d does not indicate or point to the HAZ zone.

11. Page 8, 252nd to 258 lines: this paragraph should be rewritten because the word “respectively” appears 6 times very close to each other.

12. Page 12, 359th line: Change the words SZ area for SZA and SZ volume for SZV

Author Response

This article reports the evaluation of RFSSW joints of 5 mm and 7 mm diameter of 2A12Al/7B04Al, at different rotation speeds, dwell time and depth of penetration, analyzing the microstructure, mechanical properties, fracture behavior, for joints of 5 and 7 mm diameter; to help determine the optimal welding parameters joints. This submission seems to show interesting results for the purpose of this study, I have minor comments of this document.

Minor coments

-Throughout the writing they use the word sleeve, this word is not appropriate, it is better to change it for shoulder

Comments:

  1. Page 1, 16th line of the abstract: clarify whether the diameters of 5 and 7 mm are from the welding tool or the shoulder or SZ, it is not clear.
  2. Page 1, 17th and 18th lines: Statement with incomplete idea “The macro/microstructures of the cross-section, the geometrical characteristics of the effective welding depth (EWD), the stir zone area (SZA), and stir zone volume (SZV)”
  3. Page 2, 76th line: Include references in square brackets.
  4. Page 2, 85th line and table 1: ¿Is the composition presented in at% or wt%?
  5. Page 2, 89th line: clarify whether the diameters of 5 and 7 mm are from the welding tool or the shoulder or SZ, it is not clear. Change the word tools by tool, in singular.
  6. Page 2, 91st line: Change the word tools by tool, in singular.
  7. Page 2, 90-94th lines: this paragraph should be rewritten because the word “respectively” appears 5 times very close to each other.
  8. Page 3, first paragraph: there is no mention that in the first stage occurs initial frictional preheating and for the second stage they don't mention that at the same time that the shoulder penetrates, the stirring pin retracts to create a chamber for the displaced material to flow into.
  9. Page 4. In the figure caption the text is repeated, rewrite it.
  10. Page 8. Figure 6d does not indicate or point to the HAZ zone.
  11. Page 8, 252nd to 258 lines: this paragraph should be rewritten because the word “respectively” appears 6 times very close to each other.
  12. Page 12, 359th line: Change the words SZ area for SZA and SZ volume for SZV

Responses:

Thanks very much for such detailed suggestions. The corresponding changes have been highlighted in the text.

Reviewer 3 Report

Comments and Suggestions for Authors

The research concerns an interesting topic. Text is effective, clear and well organized but need improvements. The abstract is correct, contains a summary, key findings, but need revision. However, the  Methodology is well described but also need revision too. It is possible to reproduce the tests carried out on the basis of the article. The conclusions are questionable, may need to be changed after more accurate data analysis. Few detailed comments have been presented described as follows:

 +  The introduction section intended to show (highlight) parts of the problem that are not solved by other scientists.

+ The outcome of the review is the identification of a “gap” of research that is not occupied by other scientists in this problem in a clear way.

+ The methodology, Methods, and Materials section must give a clear overview of what was done and give enough information to replicate the study (like a recipe!); be complete, but make life easy for your reader! break into smaller sections with subheadings, cite references for commonly used methods, and display a flow diagram or data table where possible. By example, the authors mention: At the same time, the sleeve retreated at the same moving speed. Then, more information is required.

+  Key assumptions and their implications could have been elaborated

+ The authors should think over the real significance of their results and try to rewrite this section to improve understanding of the conclusions

+ I recommend that the authors discuss the results in depth for the clarity of the idea to the readers and to have a real impact of the present research.

In detail, the authors mention:

+ Optical microscope (OM, OLYMPUS-GX71) and scanning electron microscope (SEM, CIQTEK-SEM3100). What are the parameters for...

+ The specimen surface was etched using the Keller solution before observation. Observation, any mechanical failure detected? Any reference?

+ From Figure 2, the behavior identifies that more rotation speed decreases the tensile shear force. Why? What is the physical phenomenon here according to the welding parameters?

+ How many replies were developed for each test?

+ From figure 3, error bars are required. Also, description of X axis is missing.

+ The results obtained needs to be compared with other references related to this work! Only a description of the cross-sectional views is reported.

+ Figure 8 its hard to see. Improve it.

Comments on the Quality of English Language

Minor issues detected

Author Response

The research concerns an interesting topic. Text is effective, clear and well organized but need improvements. The abstract is correct, contains a summary, key findings, but need revision. However, the Methodology is well described but also need revision too. It is possible to reproduce the tests carried out on the basis of the article. The conclusions are questionable, may need to be changed after more accurate data analysis. Few detailed comments have been presented described as follows:

Comment 1:

The introduction section intended to show (highlight) parts of the problem that are not solved by other scientists.

Response 1:

Thanks very much for the suggestion. The introduction has been streamlined and revised to make it more logical. Page 1 lines 42-44. Page 2 lines 45-46, lines 48-50, and lines 52-66.

Comment 2:

The outcome of the review is the identification of a “gap” of research that is not occupied by other scientists in this problem in a clear way.

Response 2:

A summary of the problem to be solved is added to the introduction section. Page 2 lines 61-66.

Comment 3:

The methodology, Methods, and Materials section must give a clear overview of what was done and give enough information to replicate the study (like a recipe!); be complete, but make life easy for your reader! break into smaller sections with subheadings, cite references for commonly used methods, and display a flow diagram or data table where possible. By example, the authors mention: At the same time, the sleeve retreated at the same moving speed. Then, more information is required.

Response 3:

The materials and methods section added subheadings and modified the content in detail. Page 2 lines 84-88 and lines 90-93. Page 3, lines 96-97, lines 99-103, line 107, and lines 115-117. Page 4 line 119.

Comment 4:

Key assumptions and their implications could have been elaborated

Response 4:

Detailed instructions have been mentioned in the article.

Comment 5:

The authors should think over the real significance of their results and try to rewrite this section to improve understanding of the conclusions

Response 5:

We have modified the results and discussions section. Page 4 lines 122-123, lines 127-128, and lines 133-134. Page 5 lines 146-147 and lines 155-158. Page 6 lines 164-166, lines 169-170, and lines 192-193. Page 7 line 200. Page 8 lines 214-216, lines 219-220, lines 223-224, lines 228-229, line 231, lines 233-234, and lines 236-239. Page 9 lines 241-243, lines 246-250, line 252, and lines 270-274. Page 11 lines 308-309 and lines 311-313.

Comment 6:

I recommend that the authors discuss the results in depth for the clarity of the idea to the readers and to have a real impact of the present research.

Response 6:

We have modified the results and discussions section. Page 4 lines 122-123, lines 127-128, and lines 133-134. Page 5 lines 146-147 and lines 155-158. Page 6 lines 164-166, lines 169-170, and lines 192-193. Page 7 line 200. Page 8 lines 214-216, lines 219-220, lines 223-224, lines 228-229, line 231, lines 233-234, and lines 236-239. Page 9 lines 241-243, lines 246-250, line 252, and lines 270-274. Page 11 lines 308-309 and lines 311-313.

In detail, the authors mention:

Comment 7:

Optical microscope (OM, OLYMPUS-GX71) and scanning electron microscope (SEM, CIQTEK-SEM3100). What are the parameters for...

Response 7:

The optical microscope is a bright field image with a 50x objective. SEM images are bright field images at 20kV acceleration voltage.

Comment 8:

The specimen surface was etched using the Keller solution before observation. Observation, any mechanical failure detected? Any reference?

Response 8:

The mechanical test samples did not etch. OM observed prior to the polishing etch.

Comment 9:

From Figure 2, the behavior identifies that more rotation speed decreases the tensile shear force. Why? What is the physical phenomenon here according to the welding parameters?

Response 9:

In Fig. 2a, the tensile-shear force of the joints is simultaneously affected by the tool rotation speed, the plunge depth and the dwell time. Under the condition of the same rotation speed, the deeper the plunge depth is, the better the mechanical properties are. This is because the more sufficient the flow of materials in the stirring zone, the better the metallurgical bonding of aluminum alloy. Under the condition of the same plunge depth, the higher the speed is, the better the metallurgical bonding is, but because of the influence of dwell time, there is a difference in energy input, which ultimately leads to the difference in mechanical properties.

Fig. 2b shows the effect of welding diameter on the mechanical properties of the joints. The effect of tool rotation speed, plunge depth and dwell time on different diameter joints is opposite.

Comment 10:

How many replies were developed for each test?

Response 10:

Three samples were selected for each parameter of the joints for mechanical property testing. Page 3 lines 115-117

Comment 11:

From figure 3, error bars are required. Also, description of X axis is missing.

Response 11:

Fig. 3 shows the effect and benefit diagram of three factors, namely, rotation speed, plunge depth, and dwell time, calculated by Minitab software according to orthogonal experiment results.  The horizontal axis is the three influencing factors, and the vertical axis is the benefit statistical results without error bars.

Comment 12:

The results obtained needs to be compared with other references related to this work! Only a description of the cross-sectional views is reported.

Response 12:

The mechanical properties of 2-series and 7-series aluminum alloy refill friction stir spot welding joints are affected not only by welding parameters but also by the thickness of aluminum alloy plate. In this paper, a 1.5mm thick 2A12Al plate and a 2mm thick 7B04Al plate are used for welding, and there are few researches on the welding of this system.

Comment 13:

Figure 8 its hard to see. Improve it.

Response 13:

The figure has been improved.

Reviewer 4 Report

Comments and Suggestions for Authors

The manuscript named Microstructure and mechanical properties of refill friction stir spot welded joints of 2A12Al and 7B04Al: Effects of tool size and welding parameters describes the refilled friction stir spot welding method as a strategy to solve dissimilar light metal joints. During the welding process three zones were evaluated the effective welding depth (EWD), the stir zone area (SZA), and the stir zone volume (SZV).

This paper is quite interesting and relevant for welding aluminum. In general, the article is well written describing the problem statement and the state of the art. The results are properly discussed, using support figures and tables.

I just have two suggestions to improve the document.

1.- Provide a brief description of the method or a book reference of the Taguchi experimental design method.

2.- Include in the document the software used to develop the Taguchi experimental design.

Author Response

The manuscript named Microstructure and mechanical properties of refill friction stir spot welded joints of 2A12Al and 7B04Al: Effects of tool size and welding parameters describes the refilled friction stir spot welding method as a strategy to solve dissimilar light metal joints. During the welding process three zones were evaluated the effective welding depth (EWD), the stir zone area (SZA), and the stir zone volume (SZV).

This paper is quite interesting and relevant for welding aluminum. In general, the article is well written describing the problem statement and the state of the art. The results are properly discussed, using support figures and tables.

I just have two suggestions to improve the document.

Comment 1:

Provide a brief description of the method or a book reference of the Taguchi experimental design method.

Response 1:

Thanks very much for the suggestion. We cited two articles about the Taguchi experiment. Page 2 line 92. Taguchi method is a scientific method to study and deal with multi-factor experiments, and it is a large branch of mathematical statistics.  The normalized-orthogonal table is used to select the test conditions scientifically and arrange the experiment reasonably.  The main advantage is that a few representative test schemes can be selected among many test schemes, and the optimal scheme can be inferred through the analysis of the results of these test schemes.  At the same time, further analysis can be made to obtain more information about various factors than the test results themselves.

A book reference of the Taguchi experimental design method with the book title “Taguchi's Quality Engineering Handbook” is supplemented.

Comment 2:

Include in the document the software used to develop the Taguchi experimental design.

Response 2:

The Minitab18 software was added to the article. Page 2 line 92.

Reviewer 5 Report

Comments and Suggestions for Authors

The manuscript deals with an interesting research topic, having a sound methodology  and insightful technological findings. In this respect the manuscript can be accepted for publication at the Materials journal after the consideration of the following comments suggested.

1. In the Abstract section authors can provide specific numerical data that better describe the conditions under which the effectiveness of EWD, SZA, SZV, were measured. This numerical data will provide a more comprehensive understanding of the test parameters followed in the experimental session of the analysis: “.….the effective welding depth (EWD), the stir zone area (SZA), and stir zone volume (SZV).

2. The examination of joining aluminum alloys under dissimilarity cases and the mechanical properties tested could be also affected under thermal and weather/climatic conditions. For this the reported results of EWD, SZA, SZV testing can be also succinctly discussed in section 3 under a) temperature, b) weather/moisture/water, c) sunlight/solar, conditions, if applicable. These parameters can provide a more integrated approach of the conducted analysis.

3. The mechanical properties and the technological dimension of the analysis have been satisfactorily conveyed. In this respect, at the end of section 3 authors can provide a more general overview and enrich their discussion beyond the specific conditions of: “5 mm and 7 mm diameter RFSSW joints of 2A12Al/7B04Al”. For instance, the roles of: other diameters’ available, other classifications of joints, specific energy needs required at the assembly fabrication-line could better convey those testing behaviour and wider conclusions of generalized truth and applicability, referring to the:

-optimum welding parameters

-tensile shear tests and failure microstructures

The critical point here is authors, based on their analysis and findings, to convey wider topics related to real world and large scale applications. Besides, the broader environmental concerns, technical constraints, fabrication and assembly complexity, can be considered for similar technological applications and large scale manufacturing settings globally. For this, 3-4 and cross cited paragraphs of this new subsection are recommended.

Author Response

The manuscript deals with an interesting research topic, having a sound methodology and insightful technological findings. In this respect the manuscript can be accepted for publication at the Materials journal after the consideration of the following comments suggested.

Comment 1:

In the Abstract section authors can provide specific numerical data that better describe the conditions under which the effectiveness of EWD, SZA, SZV, were measured. This numerical data will provide a more comprehensive understanding of the test parameters followed in the experimental session of the analysis: “.….the effective welding depth (EWD), the stir zone area (SZA), and stir zone volume (SZV)”.

Response 1:

Thanks very much for the suggestion. The related sentences were revised. Page 1 lines 16-18.

Comment 2:

The examination of joining aluminum alloys under dissimilarity cases and the mechanical properties tested could be also affected under thermal and weather/climatic conditions. For this the reported results of EWD, SZA, SZV testing can be also succinctly discussed in section 3 under a) temperature, b) weather/moisture/water, c) sunlight/solar, conditions, if applicable. These parameters can provide a more integrated approach of the conducted analysis.

Response 2:

Thank you very much for your suggestion. As you mentioned, the thermal and weather/climatic conditions indeed affect the mechanical properties to some extent especially for some fusion welding processes, however, as for the solid-state process of RFSSW, the welding conditions such as environment temperature and moisture have few effects, on the mechanical properties, except for some special extreme conditions. It’s a new research field to explore the affection on the mechanical properties of RFSSW joints welded under special extreme conditions. In this article, we just study the RFSSW joints welded under ordinary environmental conditions.

Comment 3:

The mechanical properties and the technological dimension of the analysis have been satisfactorily conveyed. In this respect, at the end of section 3 authors can provide a more general overview and enrich their discussion beyond the specific conditions of: “5 mm and 7 mm diameter RFSSW joints of 2A12Al/7B04Al”. For instance, the roles of: other diameters’ available, other classifications of joints, specific energy needs required at the assembly fabrication-line could better convey those testing behaviour and wider conclusions of generalized truth and applicability, referring to the:

-optimum welding parameters

-tensile shear tests and failure microstructures

The critical point here is authors, based on their analysis and findings, to convey wider topics related to real world and large scale applications. Besides, the broader environmental concerns, technical constraints, fabrication and assembly complexity, can be considered for similar technological applications and large scale manufacturing settings globally. For this, 3-4 and cross cited paragraphs of this new subsection are recommended.

Response 3:

Thank you very much for your constructive comments. The discussions and conclusions have been rewritten. Page 4 lines 122-123, lines 127-128, and lines 133-134. Page 5 lines 146-147 and lines 155-158. Page 6 lines 164-166, lines 169-170, and lines 192-193. Page 7 line 200. Page 8 lines 214-216, lines 219-220, lines 223-224, lines 228-229, line 231, lines 233-234, and lines 236-239. Page 9 lines 241-243, lines 246-250, line 252, and lines 270-274. Page 11 lines 308-309 and lines 311-313. Page 12 lines 338-349.

The joint diameter has a significant impact on the mechanical stability and fracture mode of the dissimilar aluminum alloy joint, so the conclusion of this paper was only based on the analysis of existing data. For the mechanical and fracture behavior analysis of other diameter joints, we will continue to conduct experiments and studies to support the optimal joint diameter and corresponding welding parameters.